# Biomarkers Associated with Lymph Nodal Metastasis in Endometrioid Endometrial Carcinoma

**DOI:** 10.3390/cancers14092188

**Published:** 2022-04-27

**Authors:** Mathilde Mairé, Aurélien Bourdon, Isabelle Soubeyran, Carlo Lucchesi, Frédéric Guyon, Guillaume Babin, Anne Floquet, Adeline Petit, Jessica Baud, Valérie Velasco, Denis Querleu, Sabrina Croce

**Affiliations:** 1Department of Surgery, Institut Bergonie, 33076 Bordeaux, France; f.guyon@bordeaux.unicancer.fr (F.G.); g.babin@bordeaux.unicancer.fr (G.B.); 2Department of Bioinformatics, Institut Bergonie, 33076 Bordeaux, France; a.bourdon@bordeaux.unicancer.fr (A.B.); carlo.lucchesi.lc@gmail.com (C.L.); 3Department of Biopathology, Institut Bergonie, 33076 Bordeaux, France; i.soubeyran@bordeaux.unicancer.fr (I.S.); v.velasco@bordeaux.unicancer.fr (V.V.); s.croce@bordeaux.unicancer.fr (S.C.); 4Department of Oncology, Institut Bergonie, 33076 Bordeaux, France; a.floquet@bordeaux.unicancer.fr; 5Department of Radiotherapy, Institut Bergonie, 33076 Bordeaux, France; a.petit@bordeaux.unicancer.fr; 6Department of Life and Health Sciences, Université de Bordeaux, 146 rue Léo Saignat, 33000 Bordeaux, France; j.massiere@bordeaux.unicancer.fr; 7INSERM U1218, Biopathology Department, Institut Bergonie, 33076 Bordeaux, France; 8Department of Gynecologic Oncology, Agostino Gemelli University Hospital, 00168 Rome, Italy; denis.querleu@esgo.org; 9Gynecology Department, Hôpital de Hautepierre, 67200 Strasbourg, France

**Keywords:** endometrial cancer, lymph node metastasis, RNA sequencing, prediction model

## Abstract

**Simple Summary:**

In endometrial cancer, lymph node invasion assessed through surgical lymphadenectomy or sentinel lymph node biopsy is a determinant factor for the prognosis and planification of adjuvant treatment. Those surgical procedures are associated with short- and long-term complications. Recent advances in molecular characterization of endometrial cancer have provided important insights into the biological nature of tumors but have not improved the pre-operative prediction of LND. This study is a description of the transcriptomic landscape associated with lymph node metastases in endometroid endometrial carcinomas. A 54-genes expression signature was generated at analysis of the primary tumor. Differential gene expression was found between patients with and without lymph node metastasis, with an 87% accuracy. Our findings provide a basis for the development of a gene expression-based signature that can be used to pre-operatively select patients for whom surgical assessment of lymph node status is of little value, and, consequently, an unfavorable risk–benefit balance.

**Abstract:**

Introduction. Lymph node metastasis is determinant in the prognosis and treatment of endometrioid endometrial cancer (EEC) but the risk–benefit balance of surgical lymph node staging remains controversial. Objective. Describe the pathways associated with lymph node metastases in EEC detected by whole RNA sequencing. Methods. RNA-sequencing was performed on a retrospective series of 30 non-metastatic EEC. N+ and N− patients were matched for tumoral size, tumoral grade and myometrial invasion. Results. Twenty-eight EECs were analyzable (16 N+ and 12 N−). Bioinformatics Unsupervised analysis revealed three patterns of expression, enriched in N+, mix of N+/N− and enriched in N−, respectively. The cluster with only N+ patient overexpressed extra cellular matrix, epithelial to mesenchymal and smooth muscle contraction pathways with respect to the N− profile. Differential expression analysis between N+ and N− was used to generate a 54-genes signature with an 87% accuracy. Conclusion. RNA-expression analysis provides a basis to develop a gene expression-based signature that could pre-operatively predict lymph node invasion.

## 1. Introduction

Endometrial cancer is the most common gynecological tumor in developed countries, and was the fourth most frequently diagnosed neoplasm in European women in 2018 (122,000 cases; 6.6%) [1]. Although in the majority of the cases, the cancer is diagnosed at an early stage and shows a favorable prognosis, 25% of the patients present with stage III or IV disease and impaired survival [2].

The mainstay of management of endometrial cancer is hysterectomy and bilateral salpingo-oophorectomy with or without lymph node staging depending on the pre-operative risk classification [3]. According to recently published European guidelines, the type of lymph node dissection (sentinel lymph node biopsy, pelvic lymphadenectomy and para-aortic lymphadenectomy) depends on the pre-operative risk assessment. No evidence of benefit in terms of survival has yet been associated with systematic lymphadenectomy [4] but knowledge of pathological lymph node status is relevant to plan adjuvant therapy. Morbidities associated with lymph node dissection include perioperative systemic morbidity and lymphoedema [5]. Although morbidity appears to be reduced with sentinel lymph node biopsy, the risk of surgical complications is still present [6].

At present, before surgery, the risk of lymphatic spread is indirectly assessed on the basis of the combination of tumor invasion depth, tumor grade, tumor size and histotype. However, preoperative workup has limitations, including interobserver variation in the assessment of histotype and histological grade, lack of accuracy of imaging techniques regarding myometrial invasion and lymph nodal metastasis [7]. Lymphovascular invasion, a strong independent pathological feature associated with lymph nodal disease, is not available before a definitive pathological examination of the hysterectomy specimen [8]. Recent advances in the molecular characterization of endometrial cancer [9] have provided important insight into the biological nature of tumors. Different diagnostic algorithms have been proposed using three immunohistochemical markers (p53, MSH6 and PMS2) and one molecular test (mutation of the exonuclease domain of *POLE*) to identify prognostic groups analogous to the TCGA molecular-based classification [10]. Thus far, molecular subgroups have been demonstrated to improve the prediction of survival outcomes but were not identified as independent predictors of stage IIIC–IV disease [11].

Hence, additional tools including highly specific and sensitive molecular biomarkers are needed to more accurately select the women for whom complete surgical staging should be performed to adapt adjuvant therapy. Whole RNA-sequencing is an essential part of high throughput sequencing, as it provides biological information of gene expression regulation. In this study, we used whole RNA sequencing to describe the transcriptomic landscape associated with lymph node metastases in endometroid endometrial carcinomas. This study is a pilot study to investigate the genes of interest in order to develop a gene signature predictive of lymph node involvement.

## 2. Materials and Methods

### 2.1. Patients Selection

We conducted a monocentric case-control study at the Institut Bergonie, Comprehensive Cancer Center, Bordeaux, France. Patients’ inclusion criteria included (1) Clinical stage I–II endometrial cancer treated between January 2010 and February 2017; (2) initial surgery performed at the Institut Bergonie, pathological tissue available and confirmed diagnosis of endometrioid endometrial carcinoma; (3) surgical staging performed with total hysterectomy, bilateral salpingo-oophorectomy and lymph node evaluation with either bilateral pelvic lymph node dissection with or without para-aortic lymphadenectomy or sentinel-node biopsy. Exclusion criteria were: non-endometrioid carcinoma, metastatic cancer (FIGO stage IV).

Information regarding the patient’s characteristics, pre-operative tumor characteristics, type of surgery, definitive tumoral staging, adjuvant therapy and outcomes was collected retrospectively from medical records and entered into a REDCAP database after anonymization.

### 2.2. Histopathological Assessment and Molecular Features

All slides of selected patients were reviewed by a gynecopathologist (SC) to confirm the histological subtype, the tumor grade, the presence of lymphovascular invasion, stromal reaction, MELF pattern and the depth of invasion (according to the fifth edition of *Female Genital Tumors* [12]). Substantial lymphovascular invasion (LVSI) was defined by the presence of tumor cells in five or more lymphovascular spaces [3]. For each patient, the tumoral area was selected for RNA-extraction by macro-dissection.

Immunohistochemistry (IHC) was performed on paraffin-embedded (FFPE) tissue samples using the following antibodies: MLH1, MSH2, PMS2, MSH6 (clones M1, G219-1129, A16-4,SP93; Roche Diagnostics Gmbh, D-68305 Mannheim, Germany) and p53 (clone DO7, Dako, Glostrup, Denmark) to assess the MMR protein status and p53 expression [13,14]. MMR protein status was considered deficient (MMR-D) when a complete loss of nuclear expression was observed in carcinoma cells of one or more MMR protein (MLH1, MSH2, MSH6, PMS2). A weak/patchy/cytoplasmic/punctate or dot-like nuclear pattern was considered as abnormal MMR expression [15]. Particular attention was reserved for the subclonal/heterogeneous pattern of MMR staining abnormality. Aberrant p53 staining (p53abn) was defined as: (1) in case of strong and diffuse nuclear staining (p53 strong); (2) or completely negative staining (P53 absent) in the presence of internal positive control; (3) cytoplasmic and nuclear pattern in carcinoma cells [13]. As the positive internal control the normal myometrium or normal endometrium was used. As external control a case of serous carcinoma with *TP53* mutation with overexpression and a case of serous carcinoma with a *TP53* mutation with p53 absent and a normal tonsilla were used. Because the *POLE* mutation screening was not available in routine practice at the time of the diagnosis, we screened for 11 known *POLE* hotspots mutation by RNA-sequencing (on cDNA) [16].

Patients were classified into two groups according to lymph node involvement. In order to avoid any bias for the known factors associated with lymph node metastasis and available before surgery, each patient with confirmed lymph node metastasis was matched to a control patient without lymph node metastases with similar definitive tumoral size (0–20 mm, 20–35 mm and over 35 mm), tumoral grade and myometrial invasion (less than 50% or over 50%). Definitive tumoral size was available from the initial pathological record, and the tumoral grade and myometrial invasion were issued for pathological review. Because it is usually non-available before surgery, lymphovascular invasion was not used in the matching protocol.

### 2.3. Matching and Initial Statistical Analysis

All initial statistical analysis was performed on SAS Pro and R version 3.6.1. (Packages epiDisplay, prettyR, epiR, gtsummary, dplyr, survival). Patients were matched 1:1 between N+ and N− according to definitive tumoral size (0–20 mm, 20–35 mm and over 35 mm), tumoral grade (FIGO 1, 2 or 3) and myometrial invasion (less than 50% or over 50%). For each case, only two to three control cases were available. Three samples with lymph node metastases had no matching pair found in the control group but were conserved for RNA extraction. Quantitative variables are presented as mean with standard deviation and were compared using *t*-test. Qualitative variables are presented as percentages and compared using Chi-2 or Fisher’s test.

### 2.4. RNA Sequencing

Formalin-fixed and paraffin-embedded (FFPE) tissue samples of primary tumor, selected by the pathologist, were used for whole RNA-sequencing. All RNA extraction, sequencing and data analysis were performed at the Institut Bergonie, Bordeaux. Extraction of RNA was realized with the Maxwell^®^ RSC RNA FFPE Kit (Promega). Library preparations were performed according to the True Seq RNA Exome Library Prep Guide, Illumina, and then sequenced (2 × 75 bp, paired-end) in a Nextseq 500 sequencer, Illumina. All analysis was performed at the Institut Bergonie.

### 2.5. NGS RNAseq Sequence Alignment and Quality Control Pipeline

Raw RNAseq sequences were controlled for quality using a set of published tools to produce curated reads. Firstly, reads with low quality bases at 5′ and 3′ were trimmed using the Sickle package (Phred cut off 20, max trim size 30 nc) [17]. The SeqPrep package was used to remove sequencing adaptors from raw reads [18]. This package also detected an important proportion of RNA fragments whose R1 and R2 paired-end reads were overlapping and merged them into single-end reads. To keep exploiting those fragments, a home-made python script was developed that split those merged reads into new non-overlapping R1 and R2 paired-end reads. Curated reads were aligned using TOPHAT2 (based on BOWTIE2) on both the UCSC hg19 reference genome and transcriptome [19]. Finally, we applied a post-alignment quality control of aligned reads by removing reads with mapping scores lower than 20 using Samtools [20]. PCR duplicate reads were identified and removed using Picard MarkDuplicates (https://broadinstitute.github.io, accessed on 24 January 2017). Read counts were enumerated using HTSeq [21].

In order to avoid normalization bias between samples, due to an imbalanced number of genes with zero aligned sequences per sample, samples were included in analysis if and only if their total count of aligned sequences was greater than 5 M PE and the number of genes with zero counts in the sample was lower than 3500 (Appendix A).

### 2.6. Differential Analysis

Transcript count data were normalized according the VOOM method, which transforms raw count values to log2-counts per million (logCPM), estimates the mean-variance relationship and uses this relationship to compute appropriate observational-level weights [22]. The RNAseq differential gene expression between groups of samples was performed using the statistical *t*-test from the R package LIMMA, which calculates fold changes and nominal *p*-values related to each gene starting from raw expression values and the normalization weights produced by VOOM [23]. The set of nominal *p*-values from each test were adjusted according to the Benjamini–Hochberg adjustment [24]. We defined the significantly up- or downregulated transcripts using an FDR threshold of 0.05. The fold-change used to further filter the differential gene expression was set to a minimum value of 2.

To obtain discriminant signature of gene expression between two groups of samples we used the shrunken centroids method (pamr R package) from Tibshirani and et al. [25] that provides an estimate of the misclassification error of the signature by splitting the data into training ad validation sets using cross-validation.

To challenge the performance measured by the pamr signature, a homemade performance visualization method was used (Centroid Validator). Briefly, this method takes in as input the expression data set of the samples used during training, the expression data set of the validation samples, the list of genes that represent the signature and the labels for each training and validation sample indicating their membership group (N+ and N− in our study). Using the training data set, the method calculates the centroids of each group, represented by the mean expression of each gene in the signature for each group of samples. A score is calculated to predict the classification of a validation sample to a specific group. The score of a given validation sample is calculated by the distance, in terms of the Mean Square Error (MSE), between the value of each gene in the validation sample and the value of the centroid for the same gene in each group in the training data. The statistical significance of the score is assessed by randomization of the MSE. Finally, a confusion matrix between the predicted and expected classification of the validation samples is created to provide an additional measure of performance of the gene expression signature.

### 2.7. Gene Set-Enrichment Analysis

MSigDB [26] was used to identify pathways or gene ontologies in which the genes of an identified group were enriched. Oncogene and Tumor Suppressor Gene status (approximately 800 known genes) was assigned according to the annotation in the Cancer Gene Panel [27,28].

### 2.8. Mutations Analysis

Production of raw genetic alteration files (.vcf/.bcf) was performed via Samtools/Bcftools with a quality cut-off per base of Phred 20. Calculation of the number of aligned reads covering each exon was performed via HtSeq. Quality reports of raw reads were built with FastQC (FastQC—https://www.bioinformatics.babraham.ac.uk/projects/fastqc/, access on 27 January 2017). Annotation of SNV and InDel genetic variants was reported by Annovar [29].

### 2.9. Unsupervised Clustering

Unsupervised clustering analysis of RNA-Sequencing expression data was performed using agglomerative hierarchical clustering with distance criteria (1-Pearson_Correlation) and linkage criteria equal to average via the function hclust available in R [30]. We performed this procedure iteratively on a subset of genes with increasing variability based on their standard deviation. The choice of the number of genes to use for freezing the clustering configuration was determined empirically, as follows. For each clustering performed at a given standard deviation cut-off, we measured the anti-correlation between the two groups placed at the highest level in the clustering hierarchy. We assumed that clustering with anti-correlations close to 0 were associated with grouping by chance between samples. We selected the SD cutoff for which sample anti-correlation was at the lowest negative level whilst the number of genes left was no less than 3000. This procedure was meant to remove genes with weak variability that could bias clustering via spurious random correlations and, at the same time, assuring that sample grouping was determined by the largest as possible number of genes. To assess the robustness of clustering results at a given SD threshold, we performed consensus clustering via the package ConsensusClusteringPlus available in BioConductor [31]. Consensus was established after 10,000 iterations on the subset of samples obtained by leave-n-out of 40% of samples.

## 3. Results

### 3.1. Patients and Clinical Characteristics

Of the 380 patients with non-metastatic endometrial cancer presented in our multidisciplinary tumor board between 2010 and 2017, 290 had endometrioid endometrial carcinoma (EEC). Thirty-five patients (12.1%) had lymph node metastasis (N+) and among them, 17 had surgery in our center. They were paired with a control group of thirty-nine lymph node negative patients (N−), who had their primary surgery within our center as well, including a lymph node dissection. Three samples with lymph node involvement could not be paired to a control but were included in the final population (Figure 1).

Patients with (N+) and without (N−) lymph node metastases were not different for age at diagnosis, body mass index, menopausal status and smoking status (Table 1). Four patients (33.3%) in the N− group and 10 (62.5%) in the N+ group underwent pelvic and para-aortic lymphadenectomy, and seven (58.9%) patients underwent isolated pelvic lymphadenectomy in the N− group and three (18.8%) in the N+ group. The reasons for omitting paraaortic lymphadenectomy were comorbidity and morbid obesity. An average of 15 pelvic lymph nodes and 18 para-aortic lymph nodes were removed in the population, and laparoscopic surgery was the preferred technique.

### 3.2. Patients’ Histological and Molecular Classification

Mean tumor size was 46.8 mm in the N− group and 50.1 mm in the N+ group. A total of 20 EEC patients (71.4% of the overall cohort) had high histological grade and 18 (64.3%) had more than 50% of myometrial invasion (Table 2 and Appendix A). Angioinvasion was more frequent when lymph node metastases were present: nine patients (56.2%) in the N+ patients had substantial lymphovascular invasion versus only two patients (16.7%) in the N− group (*p* = 0.013). The MELF pattern was also more often described in the N+ group (62.5% versus 25.0% in the N− group) but this difference was non-significant (*p* = 0.07). Stromal invasion, peri nervous invasion and inflammatory infiltration were not different between the two groups.

No pathogenic POLE mutation was found in our study population. Thirteen EEC patients were characterized by defective MMR and classified as hypermutated: four (33.3%) in the N− group and nine (56.2%) in the N+ group. TP53 pathogenic mutations were identified in three EEC, two of them had abnormal p53 staining pattern (two were absent and one overexpressed) and one a normal wild-type staining pattern. The three EEC with TP53 mutation were N+ patients. Molecular classification distribution between the N+ and N− patients was not statistically different (*p* = 0.06), hence it did not have a predictive value.

We searched for pathogenic mutation of CTNNB1, a predictor of poor prognostic in endometrial cancer [32]. Pathogenic mutation of CTNNB1 was identified in five (41.7%) N− patients and two (12.5%) N+ patients and was not statistically different between the two groups.

### 3.3. RNA Extraction, Sequencing, Bioinformatics Quality Control

RNA extraction was performed for 31 tumor samples (14 N− and 17 N+). After bioinformatics quality control, three samples were not interpretable because of lack of a sufficient gene coverage and were excluded from the analysis.

### 3.4. Differential Gene Expression (DGE) Analysis between N+/N− Samples

In order to identify differences of expression between the two major clinical groups, we performed differential expression analysis between the 16 N+ and 12 N−.

Eleven genes were significantly different between the two groups, of which WTIP, FIGN, PRX, AVR1A, LTBP3, ASPN, EFEMP1 and MGP were upregulated in N+ patients and LIN28B, C1orf64 which were upregulated in the N− patients. However, after applying a machine learning method, no gene was found to be discriminant between the two groups (See Material and Methods-Differential analysis). Hence, this first analysis was not conclusive.

### 3.5. Unsupervised Analysis

In order to check if other molecular factors associated with gene expression regulation might be the cause of the limited results of the DGE analysis between the major clinical classes, we applied an unsupervised consensus and hierarchical clustering of RNA-sequencing data. This method allows the classification of samples only, based on gene expression data without considering the a priori N+/N− classification. This method identified three groups of patients with associated gene-clusters (Figure 2). Cluster A included 7 samples all with lymph node metastases, cluster B included 10 samples, 4 N− and 6 N+ and cluster C included 11 samples, 3 N+ and 8 N−. For the following analysis, cluster A is considered as N+ cluster, cluster C as N− and cluster B as mixed.

Apart from the lymph node invasion, no other histopathological characteristic was found statistically different between the three clusters (Appendix A). We then performed DGE analysis between the three groups, which identified 1491 genes between A and B (Appendix A), 1416 genes between A and C (Appendix A), 956 genes between B and C (Appendix A) and 1348 genes between A and B + C (Appendix A). These results show that, once the initial N+/N− cohort has been reorganized by regrouping samples which have mixed N+/N− characteristics, as the B group, then strong gene expression differences could be identified.

### 3.6. Pathway Analysis

In order to identify the biological pathway characterizing the groups A, B and C we performed Gene Set Enrichment Analysis (GSEA) (17) (See Material and Methods). Cluster A (only N+ patients) was mainly enriched in genes from Extra Cellular Matrix (ECM), Epithelial to Mesenchymal (EMT) and Smooth Muscle Contraction (SMC) (Figure 3).

In the EMT pathway, we found that MSX1 and LAMA1 genes were downregulated and SFR4, DCN, CXCL2, FAP genes were upregulated in the N− patients. In the ECM pathway, we identified that the MUC2 gene was downregulated, and COL4A3, SGCA, COL4A4 TNXB, MGP, EFEMP1 genes were upregulated in the N+ patients. In the Smooth Muscle Contraction pathway, MYH11, LMOD1, ACTG2, ACTA2 belonged to the set of genes upregulated in the N+ samples. L1CAM (L1 cell adhesion molecule), which has already been described as predictive of worse outcomes in endometrial carcinoma [33], was upregulated in cluster A (N+ patients) with a fold change of 7.8.

Two pathways were upregulated in cluster C (enriched in N−): the Cell Proliferation and Beta Cells pathways. Cluster B (a mix of N+ and N− samples) broadly overexpressed all Immune Cell types (See Material and Methods) as well as the Cell Death pathway (Figure 3).

### 3.7. Potential Gene Signature

Because the interest was in comparing groups only enriched in N+ and N−, DGE was performed between group A (N+) and C (N−). In cluster C, we compared the three N+ patients with the rest of the group (Appendix A). The three N+ patients in cluster C were classified hypermutated according to TCGA molecular classification. Those three patients were different from the eight others in terms of MELF pattern, angioinvasion, myometrial invasion and TCGA molecular classification. To generate an accurate molecular signature and compare the N+ and N− homogeneous groups, we removed three N+ samples from group C to define the group C’ containing only N− samples.

DGE analysis performed between A (N+) and C’ (N−) groups identified 1047 differentially expressed genes (Appendix A).

Using a Machine Learning method (See Material and Methods), a gene expression signature of 54 genes was found (Appendix A), which could discriminate the N+ and N− samples with a cross validated accuracy of 87%, a specificity of 86% and a sensitivity of 88% (Appendix A) [25]. Three of the genes in the signature were those associated to the EMT, nine genes to the ECM and five genes to the SMC pathways.

## 4. Discussion

Surgical assessment of lymph node involvement is an essential parameter to guide the therapeutic strategy in endometrial cancer. The sentinel lymph node (SLN) biopsy has emerged as an alternative to complete lymphadenectomy, with an overall detection rate over 80% and a specificity of 100% [34]. Although surgical complication rate appears to be reduced when SLN biopsy is performed compared to lymphadenectomy [35], they are not zero [36]. Lymph node dissection prolongs the duration of surgery, the risk of pre- and post-operative complications and may be difficult to complete in obese patients with prior surgeries or patients with major co-morbidities.

We performed a genome-wide analysis in order to describe the molecular landscape associated with lymph node metastases. RNA-sequencing data were comprehensively analyzed to identify genes of interest for a potential molecular signature predictive of lymph node metastasis. Unsupervised analyses found three distinct clusters, one of them was significantly enriched in lymph node involvement. Comparison of those two specific groups allowed us to identify three overexpressed pathways in patients with lymph node metastases and a potential gene signature for lymph node involvement of 54 genes with an 87% accuracy.

In the study’s population, 12% of EEC presented with lymph node metastases and the major clinical and histologic difference between the two groups was lymphovascular space invasion (LVSI). In 2017, a Cochrane database review by Frost et al. noted that lymph node metastases were found in approximately 10% of women who, before surgery, were thought to have cancer confined to the uterus [5]. Substantial LVSI is the strongest prognostic factor for lymph node recurrence [8], but is not available on initial biopsy. Another recently identified prognostic factor, MELF pattern, is not available before final pathology of the hysterectomy specimen [37].

The three overexpressed pathways in the group with lymph node involvement are EMT, ECM and SMC, and these data are consistent with the literature. The ECM networks create a microenvironment for cell growth and development, favorable for tumoral invasion [38]. EMT has been identified as a principal component of tumor metastasis, by increasing the motility and invasiveness of cancer cells [39]. The SMC pathway regroups genes that control the interaction between cancer and stromal cells, found to have an impact on cancer invasion [40]. The immune cells’ pathway was overexpressed in cluster B, and they are known to shape the immune environment in endometrial cancer and predict the prognosis [41].

In particular, the *MSX1* (EMT pathway) was found to be downregulated in EEC with lymph node metastasis. The gene *MSX1* (Msh homeobox 1) encodes the homonymous protein MSX1, a transcription repressor, that has an inhibitory effect on the cell cycle [42]. Eppich et al. found that higher *MSX1* expression (assessed by immunochemistry) correlates with improved long-term survival [42]. In a study by Yang et al. the research team developed a nine-transcription factors’ prognostic signature, and *MSX1* was one of them. However, the eight other genes from Yang’s signature were not differentially expressed in our study. Furthermore, this study evaluated gene expressions associated with prognosis but not lymph node metastases and was developed with data obtained in silico from the TCGA, with all histologic types of EC (not only endometrioid histotype, as in our series). Other differentially expressed genes found in our study were previously described as associated with tumorigenesis or poor prognosis. Fibroblast Activation Protein (*FAP*), from the EMT pathway, was significantly upregulated in EEC with lymph node metastasis. It had been described as predictive of poor prognosis in high-grade serous ovarian carcinomas [43]. *MGP* (Matrix G1a protein), from the ECM pathway, was upregulated in samples with lymph node metastasis and its role of promoting proliferation, migration and transformation processes in triple negative breast cancer was assessed by Kong et al. [44]. *ACTGA* (SMC pathway) was also upregulated in our population N+, and was identified as a promoter of cells’ migration and tumor metastasis in hepatocellular cancer [45].

Our study is focused on lymph node metastasis prediction. A few previous papers have investigated the same topic. Huang et al. established an eight-gene biomarkers panel for predicting lymph node metastasis in patients with early stage endometrial cancer [46]. They used RNA-sequencing data from TCGA and a small panel of their own patients’ samples. None of the eight genes they selected was found in our signature. Only three genes of this signature (*EYA2*, *MSX1* and *STX18)* were differentially expressed between N+ and N− in our population. In their study, samples were mixed between local patients and TCGA patients, clinical data in TCGA are missing and comparison of characteristics between the two different population is difficult. Population disparities (Asian patients in Huang et al. and European patients in our study) could account for differences with our study. A few studies tried to develop a gene signature for lymph node involvement but none found the same differentially expressed genes that we did in our study. Kang et al. in 2019 developed a 12 gene signature predicting lymph node metastasis with 100% sensibility and 41% specificity in the validation set [47]. None of those 12 genes were differentially expressed in our groups. However, no information on other clinical predictors was available in this study and the training and validation cohort had the same origin with a risk of overfitting. Other developed signatures were often developed using data from TCGA patients without information on the surgical techniques of the center, fixation’s delays and clinical data [46,48]. Those parameters (in particular the time of ischemia before the formol fixation) are critical and could dramatically affect the gene expression in endometrial cancer.

In this study, the only patients who were selected were those for whom surgery was performed in Institut Bergonie, with guaranteed homogeneity in terms of surgical technique and sample processing in pathology laboratory. Our population was uniformly treated, with complete clinical data. The diagnosis of EEC and, in particular, the evaluation of histological parameters which have risk factors’ value such as myometrial invasion, histological grade and the assessment of lymphovascular space invasion are subject to interobserver variability [49,50]. Moreover, diagnostic criteria have evolved during the selection period. Currently, lymph node invasion risk is assessed pre-operatively on the basis of tumor size, histological grade and myometrial invasion. Our study population of N+ patients was matched to N− patients with the same tumoral characteristics. This method ensures the comparability of our two study groups, to prevent confusion bias. Non-endometrioid endometrial cancers were excluded from our analysis to guarantee homogeneity between the different groups and to ensure that differential expression found between the groups was exclusively due to lymph nodal metastases.

This preliminary study to assess molecular markers of lymph node invasion focused on post-operative samples. Considering that evaluation of molecular alteration in pre-operative endometrial specimens shows a high concordance with the definitive hysterectomy specimen [51], this paves the way for a preoperative assessment of the risk of lymph node metastasis.

One of the limitations of our study is the low number of patients that would not allow us to test and validate a gene expression signature. For three N+ subjects, no control was found but extraction was performed to maximized RNA-expression data for the N+ patients. This could introduce confusion bias, however the N+ and N− groups were still comparable for tumor size, tumor grade and myometrial invasion.

Our study focused mainly on potential biomarkers and pathways’ description. Cross-validated accuracy of the gene expression signature between the enriched groups is 87%. An additional validation of this signature on an independent test set is needed before we could use them in clinical practice.

## 5. Conclusions

In this study, we described the differential gene expression related to lymph node invasion. Our description of the molecular landscape related to lymph node metastasis highlights the role of three pathways: extracellular matrix; epithelial to mesenchymal and smooth muscle contraction, and those results are consistent with literature.

This RNA-expression profile may provide a basis for further study to develop a gene expression-based signature that could pre-operatively select patients in whom surgical assessment of lymph node status has a low yield and, consequently, an unfavorable risk–benefit balance. In order to assess reproducibility of our results, we will have to test our gene signature prospectively on an independent population.

## Figures and Tables

**Figure 1 cancers-14-02188-f001:**
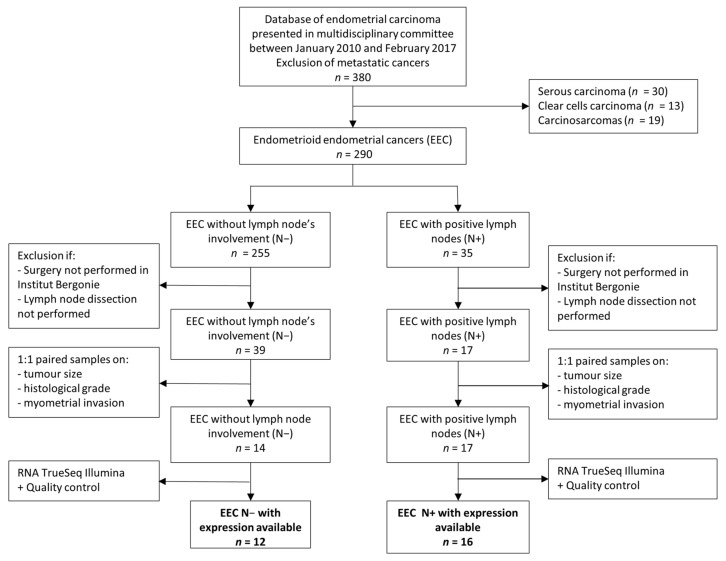
Flow chart, Institut Bergonie, France, 2010–2017.

**Figure 2 cancers-14-02188-f002:**
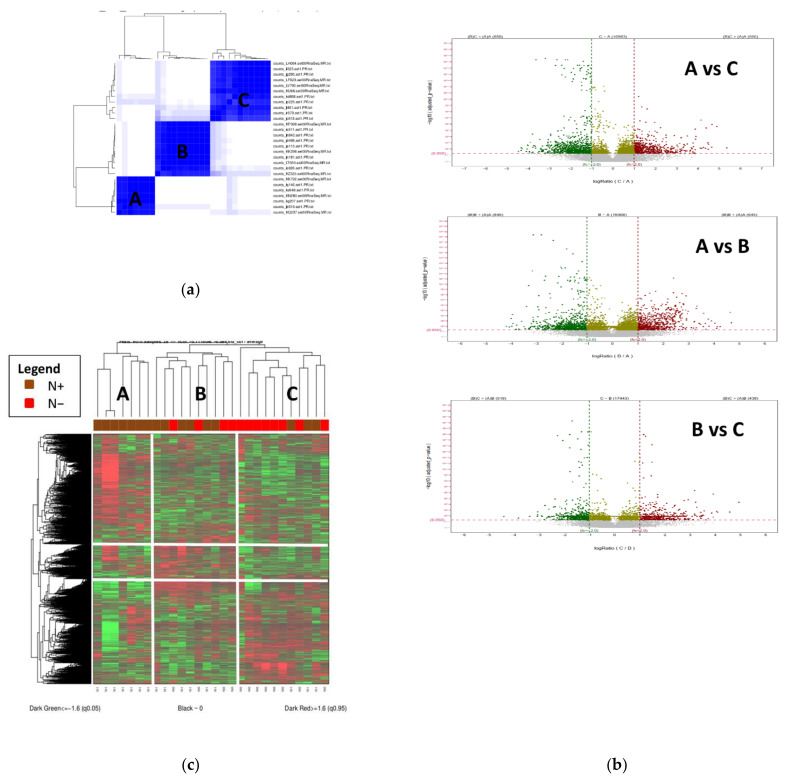
(**a**). Unsupervised analysis of our population highlights 3 groups A (N = 7), B (N (N = 10) and C (N = 11) (**b**). Heatmap of differentially expressed genes between the 3 identified clusters (A, B and C) (**c**). Volcano plots of up-regulated (red) and down-regulated (green) differentially expressed genes between the three clusters, Institut Bergonie, 2010–2017.

**Figure 3 cancers-14-02188-f003:**
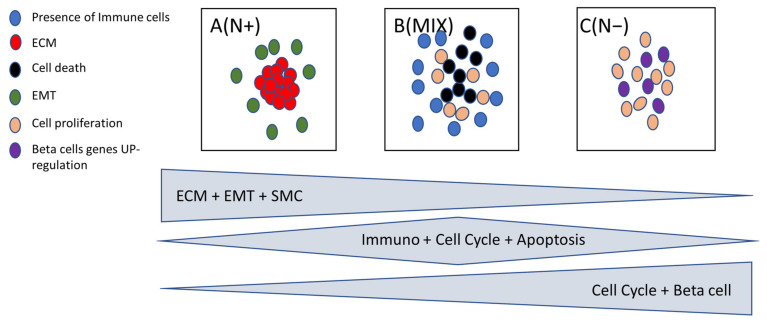
Summary pathway analysis differentially expressed between cluster A (N+), cluster B (MIX) and cluster C (N−).

**Table 1 cancers-14-02188-t001:** Clinical characteristics of EEC without lymph node involvement (*n* = 12) and with lymph node involvement (*n* = 16), Institute Bergonie, 2010–2017.

Characteristics	EEC with Negative Lymph Nodes (*n* = 12)		EEC with Positive Lymph Nodes (*n* = 16)	
	*n* (%)	Mean (SD)	*n* (%)	Mean (SD)
Patient’s characteristics				
**Age at diagnosis**		63.2 (10.7)		67.8 (6.9)
**Comorbidities**				
Cardiovascular disease	4 (33.3)		7 (43.8)	
History of cancer	2 (16.7)		2 (12.5)	
Thyroid pathology	3 (25.0)		4 (25.0)	
Chronic renal failure	1 (8.3)		0 (0.0)	
Diabetes	1 (8.3)		1 (6.2)	
Pulmonary pathology	1 (8.3)		0 (0.0)	
Neurological pathology	1 (8.3)		0 (0.0)	
**Menopaused at diagnosis**	10 (83.3)		16 (100.0)	
**Smokers**	2 (16.7)		1 (7.7)	
**Body mass index (kg/m^2^)**		26.3 (5.7)		28.2 (5.9)
**Surgical characteristics**				
**Type of initial surgery**				
Laparoscopic surgery	9 (75.0)		10 (62.5)	
Open surgery	3 (25.0)		6 (37.5)	
**Initial lymph node staging**				
Sentinel lymph node biopsy	1 (8.3)		1 (6.2)	
Pelvic and para-aortic lymphadenectomy	4 (33.3)		10 (62.5)	
Pelvic lymphadectomy alone	7 (58.3)		3 (18.8)	
No initial lymph node dissection	0 (0.0)		1 (6.2)	
**Digestive resection**	0 (0.0)		2 (12.5)	
**Omentectomy**	4 (33.3)		7 (43.8)	
**Surgical re-staging**	1 (8.3)		4 (25.0)	
**Proportion of removed pelvic lymph nodes**		14 (8.9)		16 (7.0)
**Proportion of removed para-aortic lymph nodes**		22 (8.3)		15 (11.1)

**Table 2 cancers-14-02188-t002:** Comparison of histopathological characteristics and molecular patterns of EEC without lymph node involvement (*n* = 12) and with lymph node involvement (*n* = 16), Institut Bergonie, 2010–2017.

Pathological Characteristics	EEC with Negative Lymph Nodes (*n* = 12)		EEC with Positive Lymph Nodes (*n* = 16)		
	*n* (%)	Mean (SD)	*n* (%)	Mean (SD)	*p*-Value ^1^
**Tumoral size**		46.8 (22.5)		50.1 (25.9)	0.7
**Histological grade**					0.9
Low grade (grade 1 and grade 2)	3 (25.0)		5 (31.2)		
High grade (grade 3)	9 (75.0)		11 (68.8)		
**Myometrial invasion**					0.9
≤50%	4 (33.3)		6 (37.5)		
>50%	8 (66.7)		10 (62.5)		
**Angioinvasion**					0.013
Absence	9 (75.0)		3 (18.8)		
Non-substantial	1 (8.3)		4 (25.0)		
Substantial	2 (16.7)		9 (56.2)		
**Number of angioinvasion** **(if presence of angioinvasion)**		3.9 (9.4)		7.2 (8.4)	0.3
**Stromal Reaction**					0.12
Presence	5 (41.7)		12 (75.0)		
Absence	7 (58.3)		4 (25.0)		
**MELF pattern**					0.07
Presence	3 (25.0)		10 (62.5)		
Absence	9 (75.0)		6 (37.5)		
**Inflammatory infiltration**					0.9
Presence	5 (41.7)		6 (37.5)		
Absence	7 (58.3)		10 (62.5)		
**Peri nervous invasion**					0.4
Presence	1 (8.3)		4 (25.0)		
Absence	11 (91.7)		12 (75.0)		
**Molecular classification group (TCGA)**					0.06
Ultramutated (POLE mutation)	0 (0.0)		0 (0.0)		
Hypermutated (MSI)	4 (33.3)		9 (56.2)		
Serous-like (TP53 mutation)	0 (0.0)		3 (18.8)		
Non specific molecular profile	8 (66.7)		4 (25.0)		
**CTNNB1 mutation**					0.1
Pathogenic mutation	5 (41.7)		2 (12.5)		
No pathogenic mutation	7 (58.3)		14 (87.5)		

^1^ Fisher’s exact test; Two Sample *t-*test.

## Data Availability

The datasets generated and/or analyzed in this study are available upon reasonable request.

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
