# Peer review of "Biomarkers Associated with Lymph Nodal Metastasis in Endometrioid Endometrial Carcinoma"

_cancers, 2022, doi:10.3390/cancers14092188_

Round 1
Reviewer 1 Report
My dears,
Very interesting study. I enjoyed reading it. Well structured and explained. While the number of patients are low, they are very well selected and homogenous!
I do feel like the results are a bit too abrupt. You should talk more about the N+ signature. It's the most important point of the paper.
The patients you eliminate from cluster C for your last analysis: are they different from the N- ones in terms of of the TCGA EC classification (p53, NS, MSI. I understand you don't have POLE so I'm not mentioning it)?
Did you notice any relationship between your N+ gene signature and the TCGA EC classification? Did you check for ARID1A mutations? if not I suggest you take a look since it's been associated with various aggressive features
Do you have any follow-up data in terms of recurrence? i saw that your cohort is 2010-2017.
I suggest taking a look at the TCGA UCEC expression data and check your signature. You could be able to validate them or at least a few.
Please find my other comments in the pdf file.
Best of luck in the future and may the Force be with you!!!
Yours truly,
a reviewer

Reviewer 2 Report
It is an interesting paper that is very well written. As always, validation is the most important, and I hope that we can conduct forward-looking research based on this in the future.
Author Response
Thank you for your interest in our work.
All corrections suggested were made in the final text.
English language and style error were corrected
Reviewer 3 Report
I read with great interest the Manuscript titled “Biomarkers associated with lymph nodal metastasis in Endometrioid Endometrial Carcinoma” (cancers-1663821), which falls within the aim of Cancers.
In my honest opinion, the topic is interesting enough to attract the readers’ attention. Methodology is accurate and conclusions are supported by the data analysis. Nevertheless, authors should clarify some point and improve the discussion citing relevant and novel key articles about the topic.
Authors should consider the following recommendations:
- Manuscript should be further revised by a native English speaker.
- Inclusion/exclusion criteria should be better clarified.
- The Authors did not mention the sample size calculation for their study. It is essential to specify this data in order to guarantee an adequate significance of the results obtained by the Authors.
- Discussion could benefit from a brief mention about the surgical approach for SLN biopsy according to actual state of art (see PMID: 35117375; 28410225) and the most common complications that could be seen during a minimally invasive approach (see PMID: 33820674; 31421249)
Reviewer 4 Report
The authors should better explain the purposefulness of the study beforehand. She herself admits that there are still discussions about the usefulness of iliac or periortic lymphadenectomy in relation to prolonged survival.It is necessary to clarify how the preoperative determination of lymph node involvement will change (individualize treatment) because it is missing at work. We all know that periortic and iliac lymphadenectomy prolongs the time of surgery, which is sometimes longer due to technical reasons - difficult operating conditions in obese patients. Please include more of the benefits of your research in the discussion.
It would be interesting to include in the research patients with non-endometrial cancer of the non-endometrial type, who were excluded from it. Perhaps it would be a good idea to explain more in the discussion why such a decision was made . The results presented in the table are transparent. However, it is evident that the group of patients with low grading is much smaller
Round 2
Reviewer 4 Report
the revised manuscript fully takes into account the recommendations I have suggested